# A Single-Center Retrospective Analysis of 14 Head and Neck AVMs Cases Treated with a Single-Day Combined Endovascular and Surgical Approach

**DOI:** 10.3390/jpm13050757

**Published:** 2023-04-28

**Authors:** Paolo Gennaro, Samuele Cioni, Sandra Bracco, Flavia Cascino, Linda Latini, Manfredi Abbagnato, Olindo Massarelli, Guido Gabriele

**Affiliations:** 1Maxillofacial Surgery Operative Unit, Department of Mental Health and Sense Organs, Santa Maria Le Scotte, University Hospital of Siena, 53100 Siena, Italy; 2Neuroimaging and Neurointervention Unit, Santa Maria Le Scotte, University Hospital of Siena, 53100 Siena, Italy

**Keywords:** head and neck surgery, multidisciplinary approach, oral maxillo-facial surgery

## Abstract

Arteriovenous malformations (AVMs) are rare congenital defects of vascular development whose treatment remains challenging. The paper presents a retrospective single-center study of 14 patients with AVMs of the head and neck region undergoing combined endovascular and surgical treatment in a single day. AVM architecture and therapeutic strategies were determined on the basis of angiographic studies, while the psychological involvement of each patient was assessed by means of a questionnaire. Most of the 14 patients achieved satisfactory clinical results with no recurrences, good aesthetic and functional results, and most patients reported improved quality of life. The combined endovascular and surgical approach is an effective treatment for AVMs of the head and neck and performing it on the same day is a possible option often accepted by patients which guarantees operative advantages for the surgeon.

## 1. Introduction

Arteriovenous malformations (AVMs) are rare congenital defects of vascular development accounting for 1.5 percent of all vascular anomalies [1].

AVMs have a rapid flow and are composed of a complex arterial network, called “nidus”, drained directly by one or more veins without the interposition of a capillary bed.

They can affect any type of vessel, from any region, with a preference for the head and neck area (50% of lesions are in the oral and maxillofacial region) [2].

Although they are not malignant, their behavior can be locally aggressive and destructive, leading to complications such as severe disfigurement, ulceration, massive hemorrhage, pain and, in the worst cases, heart failure.

The medical history, physical examination, and imaging features led to the diagnosis [3].

They tend to expand with occasional abrupt increases in growth, triggered by trauma, hormonal changes, or iatrogenic causes [4].

The treatment of AVMs is still controversial; there are no staging criteria or standardized guidelines, and treatment options vary from a conservative approach to more aggressive strategies.

Classical treatment includes surgical ligation [5], which is now widely discouraged as some evidence shows that this procedure triggers silent malformation with the development of collateral arterial sources that could enlarge and worsen the patient’s condition [6].

Ethanol sclerotherapy, historically the main agent for extracranial treatment [7], is used for small AVMs, although several complications have been reported, including skin necrosis and peripheral nerve lesions. These can be minimized with highly selective access to the nidus and thorough knowledge of vascular anatomy, especially in the head and neck region, where potentially very serious complications can occur due to ECA-ICA (External Carotid Artery-Internal Carotid Artery) anastomoses and ECA supply to the cranial nerves. However, completely avoiding them may be impossible if the AVM itself affects the skin or mucosal surfaces. Sudden severe cardiovascular collapse is a rare complication of ethanol sclerotherapy for vascular malformations. Its mechanism is unknown: suppressed cardiac conductivity and precapillary pulmonary artery spasm have been suggested. One case of acute lung capillitis/vasculitis has been documented [8,9]. Moreover, multiple treatment sessions may be needed depending on the volume and behavior of the lesion [10].

Endovascular embolization treatment can be performed with a trans-arterial approach [11], with a direct percutaneous puncture of the nidus [12], or with a retrograde trans-venous approach [13]. The success rate of complete closure in a single therapeutic modality is about 65% and therefore many sessions are required to ensure complete closure of the AVM, with, often, a high rate of recurrence. Complications related to embolic material can be varied; in particular, when endovascular treatment is not followed by a surgical phase, there could be recurrences after the natural degradation of embolic materials [14].

In the past, surgical excision alone was more widely used than endovascular embolization therapy, but today this option is reserved for small, localized AVMs. The risk of massive bleeding during surgical removal of an AVM is unacceptable, so this is rarely a good treatment choice [15].

A recent meta-analysis showed that, currently, there is no standardized treatment for extracranial AVMs of the head and neck and that the methodological quality of the available publications is too heterogeneous to draw conclusions on optimal treatment strategies [16].

A multidisciplinary approach with a combined treatment based on endovascular embolization and surgical excision is a good compromise and is rated a good choice by several studies [17].

The purpose of this study was to review our experience with a combined approach using endovascular embolization followed by surgical resection on the same day, showing our good results.

## 2. Materials and Methods

### 2.1. Patients

This paper proposes a retrospective single-center study of 14 patients who received combined endovascular-surgical treatment for head and neck arteriovenous malformation at the Azienda Ospedaliera Universitaria di Siena between 2015 and 2022.

We enrolled only extracranial AVMs of the head and neck, evaluated with Computed Tomography (CT) and Magnetic Resonance Imaging (MRI) to assess the extent and tissue planes involved, diagnosed with angiography, and managed on the same day with the combined endovascular and surgical treatment.

AVMs monotherapy treatment was an exclusion criterion, as were other vascular malformations treated with the combined approach, for example, venous malformations.

All cases were evaluated by a multidisciplinary team experienced in vascular malformations, consisting of the maxillofacial surgeon, the interventional neuroradiologist, the dermatologist, and when the patient was a minor, the pediatrician.

Treatment planning was discussed, and the benefit/risk ratio was explained to each patient who subsequently signed an informed consent form for participation in this study.

The study obtained Ethics Committee approval, approval number 6/2022, in accordance with the Declaration of Helsinki for biomedical research involving human subjects and with the Guidelines of Good Clinical Practice.

Eight patients were male and six were females, aged between 9 and 63 years, with a mean age of 34.5 years; five patients reported having an AVM since birth and nine patients reported adolescence as the age of onset (mean age: 17 years). All patients suffered from an AVM of the head and neck, but the specific anatomical location within this district was different.

Nine of the fourteen patients (64.29%) had not undergone previous treatment, but five patients had already tried some therapies elsewhere.

The symptomatology was heterogeneous.

To assess AVM grading, we classified each case according to the SECg staging system [18] as we considered this classification the most comprehensive among those currently available in the literature. The SECg staging system is a useful tool for classifying AVMs, defining the local extent of the disease and the structures involved (S1–S4), the endovascular architecture of the malformation (E1, E2, E3), the complications (C0–C3) and the presence or absence of AVM growth (g+, g−) (Table 1).

Retrospectively, one year after treatment, a questionnaire assessing awareness and satisfaction was administered to all patients (one questionnaire was completed by a parent of a minor patient). Patients were contacted by telephone and asked five questions regarding specific symptoms, including pain, functional impairment, cosmetic deformity, impairment of daily life, and bleeding. Patients were asked to compare their symptoms before and after the combined treatment and to rate them on a 5-point scale as follows: 1 worsening of symptoms; 2 no change in symptoms; 3 minor improvement of symptoms; 4 major improvement of symptoms; and 5 complete relief of symptoms [19].

### 2.2. Endovascular Treatment

All embolizations were performed under general anesthesia with saline flush catheters under routine heparinization on a C-arm biplane angiography guide (bi-plane GE 3131 Innova).

Percutaneous access to the femoral artery was obtained and a 4–8 Fr sheath was introduced.

Preliminary angiographic studies were performed through a 4–5 Fr guiding catheter in the common carotid artery and then in the proximal part of the external carotid artery to delineate the angioarchitecture and identify arterial connections to the intracranial circulation.

Based on the endovascular features described by the SECg, treatment was initially attempted by direct percutaneous puncture with 18–21 G needles and injection of embolic agents for E1 and E2 grade nodes. The intra-arterial approach was instead used for grade E3 lesions.

Embolization was performed using different liquid or mechanical embolic materials.

The mechanical embolic materials include particles, coils, and plugs, while the liquid embolic materials include adhesive (NBCA, N-butyl cyanoacrylate) and non-adhesive materials (EVOH, Ethylene-vinyl alcohol). Among the latter are available different commercial preparations, such as Squid, PHIL, and Onyx.

In our case series, liquid non-adhesive embolizing materials are used in six patients, but two of these cases have been supplemented with the use of particles of 100 to 500 microns (Embozene, Boston Scientific Corporation, Marlborough, MA, USA) to occlude small vessels that cannot be reached in any other ways. Particles alone were used in only two patients, liquid adhesive materials alone were used in one patient, and liquid adhesive materials supplemented with particles were used in three patients.

Furthermore, in the lower two-thirds of the face and in superficial tissue, we preferred, between the liquid adhesive and non-adhesive embolizing materials, compounds without tantalum salts to avoid iatrogenic discoloration. For example, we used PHIL.

A microcatheter was advanced onto a microfilament to a peri-nidus position. Navigation of the microcatheter was performed with a Magic 1.2F (Balt Extrusion, Montmorency, France), a Headway27, Headway21, HeadwayDuo 156 (Microvention Terumo, Tustin, CA, USA), a Marksman (Ev3 Neurovascular, Irvine, CA, USA) or a Rapidtransit microcatheter (Codman & Shurtleff, Paramount Drive Raynham, MA, USA). A Synchro-14 (Stryker Neurovascular, Fremont, CA, USA) was used as a micro-drive for the Rapid Transit, Marksman, HeadwayDuo156, Headway 27 microcatheters, a Hybrid 007D (Balt Extrusion, Montmorency, France) was used as a micro-drive for the Magic 1.2F and an ASAHI CHIKAI black 18 soft tip (Asahi Intecc Co., Aichi, Japan) was used as a microguide for the Headway 21 microcatheter.

Contrast material was injected to verify the peri-nidus localization.

At the end of each embolization, the percentage of angiographic obliteration was estimated by visual assessment of the images obtained (Table 2).

### 2.3. Surgical Treatment

After the endovascular phase, a surgical procedure was performed. The patient was conducted in the operating room under general anesthesia.

Twelve patients underwent the endovascular procedure and the surgical resection in a single day, while only two patients (patients seven and twelve in Table 1) delayed the surgical approach by a few days due to excessive edema as the AVM was in their lips; this specific location tends to produce significant swelling after percutaneous endovascular treatment, so we judged that performing the excision later would be more comfortable and safer.

Due to the variety of locations of the AVMs treated, different approaches were performed: seven patients (50%) underwent a cutaneous excision and four patients (28.57%) underwent a mucosal excision, while in three patients (21.43%), both cutaneous and mucosal surgical resections were performed.

By means of all these approaches, a meticulous exposure of the AVM, previously embolized, was obtained through a blunt dissection of adjacent anatomical structures, which were carefully preserved.

Although there were no fixed margins, a tumor-like excision was performed and in superficial lesions, resection of the overlying skin was required to avoid the bluing of the skin.

Whenever possible, minimal approaches were required to reduce visible and disfiguring scars on the exposed areas in the head and neck, but not always a first intention suture was possible to close the surgical wounds and sometimes local and locoregional flaps were used.

In our case series, in two cases, the AVM involved the ear, so it was removed and the implantation of an epithesis was necessary; in two superior lip cases, a wedge resection was performed, and a transposition flap was used to reconstruct the surgical defect. In one case of the inferior lip’s AVM, a blunt dissection through the soft tissue of the lip was performed to find the embolized AVM, and after removing it, a first-intention closure was carried out, while in the other inferior lip case, the AVM was at a very advanced stage, so a large resection was necessary and a Karapandzic flap was used for lip reconstruction. In the two frontal AVMs, an advancement flap was used to close the surgical defect, while in the four cheek AVMs, different approaches were chosen. In one of these cases, being the AVM located deep in the masseter muscle, an intraoral approach was preferred because it was more suitable to reach the mass, while in the three remaining cases, the skin approach was preferred and a cervicofacial advancement flap was harvested. Finally, the AVM located in the alar nose was removed and the surgical defect was reconstructed with a bilobed flap, while, after the removal of the superior eyelid, a primary closure was performed.

The aim of the surgical procedure is the complete resection of the embolized nidus to minimize the risk of recurrences, triggered by the iatrogenic growth of collateral vessels [20].

Thanks to the embolization of the nidus and its vascular supply, intraprocedural blood loss was minimized, so that the lesion could be easily identified and safely resected.

In fact, surgical bleeding, even massive, is one of the main complications of the surgical approach alone.

## 3. Results

The percutaneous puncture alone was scheduled in eight of the fourteen endovascular procedures, but in three cases, it was converted into intra-arterial embolization to ensure better nidus obliteration. The intra-arterial embolization was administered in another six patients, for a total of nine intra-arterial procedures out of fourteen (64.2%).

Angiographic obliteration was estimated as above 90% in eight of fourteen patients (57%), above 50% in five of fourteen (36%), and below 50% in one patient (4%).

Post-surgical complications occurred in four patients (28.57%): two patients (50%) had wound dehiscence resolved with surgical plasty, one patient (25%) had an asymmetry of the lips treated with lipofilling, and one patient (25%) complained of a cheek skin discoloration due to the ectasic vessel around the AVM treated with chemical peels or microdermabrasion.

None of the patients experienced bleeding during or after the combined procedure. Two of the 14 patients required further radicalization surgery to remove residual embolized lesions. Removal of residual embolic material should be planned after the complete resolution of surgical edema, usually after 6 months.

Follow-up was 12 months with clinical control, but severe cases were also followed up with angiographic controls.

All 14 patients answered the questionnaire comparing pain, functional impairment, aesthetic deformity, impairment of daily life, and bleeding before and after treatment.

Before treatment, fourteen patients complained of aesthetic problems; eight out of fourteen (57.14%) had complete relief of symptoms, three (21.43%) had a clearly visible improvement of their aesthetic deformities, two patients (14.29%) had little improvement, and one patient (7.14%) reported no improvement in symptoms. No patients reported a worsening of symptoms.

Pain before treatment was reported by only two of the fourteen patients who experienced complete relief after our combined procedure.

Three patients reported various grades of functional impairment due to AVMs, such as visual disturbances, breathing restriction, and chewing difficulties; after the treatment, one of these (33%) experienced complete relief of symptoms, one experienced greater improvement, and one reported worsened symptoms.

Often, as a result of exposure to AVMs of the head and neck, patients experience difficulties in socializing; six of our patients reported this type of problem and four of them (66.67%) resolved it completely after treatment. One (16.67%) improved significantly, while only one reported no change.

An important aspect was the bleeding reported by four patients in the pre-treatment period; all experienced complete relief and did not go to the emergency room after our combined treatment (Table 3).

The aesthetics-functional results of patient 2 and patient 3 are shown in Figure 1 and Figure 2, respectively.

## 4. Discussion

The International Society for the Study of Vascular Anomalies (ISSVA) has recently updated its classification of vascular anomalies, dividing them into two main groups: vascular tumors and vascular malformations.

The identification of the incidence of vascular malformations in general, and of arteriovenous forms, is difficult because of confusing terminology, particularly in the frequent failure to differentiate between hemangiomas and arteriovenous malformations.

In 1977, Kennedy et al. [21] reviewed 238 studies and evaluated the incidence of vascular malformations of all types at 1.08% on average, with a range across studies from 0.83 to 4.5%. However, this study does not allow us to determine the incidence of AVMs, which can be inferred from other analyses. According to a study by Tasnadi [22] of 3573 children and to a study by Lee [23] of 1475 cases of peripheral vascular malformations, the incidence of AVMs appears to be 12%.

Among the vascular malformations, arteriovenous malformations (AVMs) are high-flow anomalies; they may be congenital or iatrogenic or arise after trauma. Localization in the head and neck is common. Based on this anatomical location, they have unique implications both in terms of the involvement of important systems, such as the visual, digestive, and respiratory systems and in terms of the aesthetic disfigurement that can lead to a psychological problem for the patient [24].

Arteriography is the gold standard for anatomical assessment and analysis of the nidus, the site of the arteriovenous shunt [25].

Treatment remains the most debated aspect of AVM management, as there is no uniformly accepted one in the literature, but several strategies are described, which have largely evolved in recent years.

Pandelaki et al. [26] considered embolization or sclerotherapy as the first-line treatment for the management of AVMs, although they reported a high risk of high-flow hemorrhage. Complete occlusion of the nidus or fistula is the goal of endovascular therapy.

There are no universally recognized guidelines regarding the choice of embolizing material.

Often it is the expertise of the operator that makes the difference, but usually, the choice depends on the angioarchitecture of the AVM, its location, the different release mechanisms, and material properties [27]. Commonly used agents are absolute alcohol, N-butyl cyanoacrylate, part of the adhesive liquid material, and ethylene-vinyl-alcohol-copolymer, a group of non-adhesive liquid materials, while mechanical materials are increasingly less used due to their poor handling.

In our experience, supported by the scientific literature, the use of non-adhesive liquid materials is to be preferred and material based on free tantalium salts is the best choice to avoid iatrogenic discoloration, especially in superficial tissues and mucous membranes.

Penington [28] described his technique based only on the surgical phase, without pre-operative embolization, because using this aid he found no reduction in bleeding and because he felt that embolization took away the most valuable guide to locating the nidus, i.e., the blood flow through the tissue. He also assessed that the real challenge is not avoiding the risk of catastrophic bleeding but knowing where to place the excision margins to prevent a recurrence.

Several studies have proposed algorithms based on different classifications. Lam et al. [29] recommended treatment based on Cho’s angioarchitecture classification, while Yakes [30] proposed possible lines of treatment based on his classification.

Instead, we used the SECg classification proposed by Colletti et al. [18] because we believe it is the most comprehensive classification. It helps with the decision-making process by dividing AVMs into curable AVMs, curable AVMs with foreseeable serious sequelae, and incurable AVMs with absolute indications and with relative (or absent) indications.

In this study, the authors show the results obtained in 14 patients with head and neck AVMs, treated with a one-stage combined endovascular and surgical approach performed on the same day.

In the craniofacial region, the timing of surgery after embolization remains a debated point [31]. Some authors suggest that it should be performed within the next 72 h to allow the AVM to shrink in size, especially if located within vulnerable structures [32].

In our study, we demonstrate the possibility of performing the surgical procedure on the same day as the endovascular embolic procedure. This approach is not only more accepted by patients as they undergo general anesthesia but also has operative advantages.

In fact, some embolic materials have very short degradation times, and this is likely to invalidate the surgical procedure, not only because of the increased risk of bleeding but also because of the unavoidable reduced accuracy in the assessment of resection margins, increasing the risk of recurrence.

However, this approach must also be supported by the hospital’s own structural features. In fact, in our case, this procedure was made possible by the fact that the interventional neuroradiology unit is a couple of floors above the maxillofacial surgery operating room, facilitating the movements that occur when the patient is under anesthesia.

Furthermore, waiting for the edema to resolve may be counterproductive because although surgery may be easier, there is a risk of underestimating AVM and performing a partial resection that will result in recurrence over the years.

Therefore, performing surgery immediately may result in larger resections to be reasonably sure of radicality, but thanks to local, regional, or free flap [33,34,35] it is possible to reconstruct even large surgical defects.

In our case history, local and regional flaps were used to reconstruct the surgical defect, despite this, as shown by the results of the questionnaire, the results obtained were judged satisfactory by the patients themselves.

Likewise, the advantage of one-day combined treatment is that thanks to the first endovascular phase, during the surgical phase the lesion can be easily identified, and the surrounding healthy soft tissue can be preserved and vascular, nerve, or muscle structures spared.

Nevertheless, various surgical techniques can be used to repair any damage immediately [36,37].

This type of treatment could be chosen regardless of the size of the AVM. However, the authors believe that the aesthetic impairment due to the location of the AVM in the head and neck requires early treatment, regardless of the growth rate or the extent of the lesion, and propose to perform the combined approach before attempting other strategies, e.g., arterial ligation or a single embolization that could result in the growth of collateral vessels and difficult-to-manage recurrences.

The treatment of AVMs of the head and neck remains challenging. According to the international literature and the author’s experience, the combination of endovascular and surgical procedures seems to be the best option to achieve a safe and complete excision of the malformation, with a low risk of recurrence.

Performing prior endovascular treatment can optimize the surgical approach not only by reducing intra- and post-operative complications, particularly massive hemorrhage, but also by minimizing the risk of recurrences amplifying the difference between healthy and pathological tissue.

In addition, complete removal of the AVM prevents residual embolizing material from emerging from the tissues as a foreign body reaction during the postembolization phase. This eventuality, which we have faced in one patient, must be managed by bringing the patient back to the operating room to perform radicalization surgery as soon as the material is visible.

What differentiates our work from other studies available in the literature is the variability of the case series, which includes as many as 14 clinical cases of complex management in which we were able to achieve good results.

These results were possible thanks to the modern interventional radiology techniques used, which include recently introduced embolizing materials and, in most cases, percutaneous accesses rather than intra-arterial approaches, which allowed us to better control the procedure and significantly reduce the complications described in the literature.

Moreover, in our work, we not only described our successes but also reported cases of recurrence and the need for re-intervention that are not reported in many works. In fact, we have assessed that these eventualities are not to be considered complications or failures, but as possible developments in the natural history of the pathology, so it is necessary to know about them and how to deal with them.

Through our experience over the years, we have learned to manage significant complications related to this condition. This has allowed us to better select patients for treatment and improve both embolization and surgical techniques.

We believe that the best management of these patients is in referral centers where there are established diagnostic and treatment pathways. Many patients still come to our attention after being seen in many hospitals without obtaining a definitive diagnosis or treatment. This is detrimental because since AVMs are degenerative diseases, delay in diagnosis can be fatal. Being able to evaluate so many patients allows us to study this pathology and advance in diagnosis while always ensuring the most cutting-edge treatment.

The prospects are many and many promising. We have also begun to investigate alternative treatments, for example, bleomycin injection and electrochemotherapy, but the small number of cases we have treated still does not allow us to draw any firm conclusions.

Our goal is to tailor therapies to the specific patient, ensuring the best possible outcome with minimal invasiveness and morbidity.

Finally, since this is a rare condition, we believe that the creation of a network allowing healthcare providers to share information about AVM patients is desirable in order to identify the most appropriate referral centers in the area for the management and treatment of this disease.

## Figures and Tables

**Figure 1 jpm-13-00757-f001:**
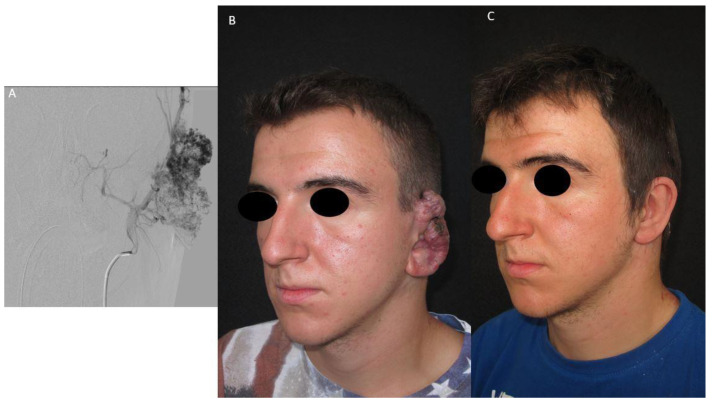
Patient 2 with a right ear AVM. He was treated with the combined approach executed on the same day, and the surgical defect was reconstructed with an epithesis. (**A**): Pre-operative angiography; (**B**): Pre-treatment photo; (**C**): Post-treatment photo.

**Figure 2 jpm-13-00757-f002:**
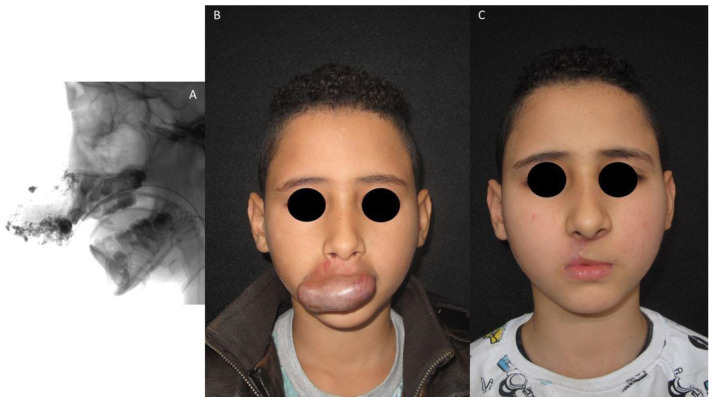
Patient 3 with a superior lip AVM. He was treated with the combined approach executed on the same day, and, during the surgical phase, a wedge resection and a transposition flap were performed. (**A**): Pre-operative angiography; (**B**): Pre-treatment photo; (**C**): Post-treatment photo.

**Table 1 jpm-13-00757-t001:** Patient Data.

SEX	AGE	ONSET	LOCATION	SYMPTOMS	PREVIOUS INTERVENTION	SECg STAGING
M	34	Birth	Left auricular cervical extending	Haemorrhage Cosmetic	Sclerotherapy	S3, E1, C3, g−
M	22	Birth	Left auricular cervical extending	Haemorrhage Cosmetic	Surgical treatment	S3, E1, C3, g−
M	9	4 yo	Superior lip	Chewing Cosmetic	Surgery ligation	S3, E2, C1, g+
M	24	15 yo	Frontal	Cosmetic	Embolization	S2, E2, C1, g+
F	40	18 yo	Right cheek	Pulsatile Cosmetic	No	S2, E1, C1, g+
M	48	23 yo	Left cheek	Chewing Cosmetic	No	S2, E3, C1, g−
F	37	22 yo	Superior lip	Chewing Cosmetic	No	S3, E2, C1, g+
M	39	Birth	Right cheek	Cosmetic	No	S3, E2, C1, g+
M	20	18 yo	Inferior lip	Cosmetic	No	S1, E2, C1, g+
F	57	25 yo	Left nose wing	Breathing Cosmetic	No	S1, E1, C1, g+
F	27	13 yo	Right cheek	Cosmetic	No	S2, E3, C1, g+
M	63	Birth	Inferior lip	Haemorrhage Cosmetic	No	S3, E2, C2, g+
F	28	15 yo	Frontal	Pulsatile Cosmetic	No	S1, E3, C1, g+
F	35	Birth	Left superior eyelid	Vision Cosmetic	Surgical treatment	S2, E3, C1, g+

**Table 2 jpm-13-00757-t002:** Endovascular Procedure’s Results.

EMBOLIC MATERIAL	TYPE OF APPROACH	ANGIOGRAPHIC OBLITERATION	EMBOLIZATION-SURGERY GAP	POST-OP COMPLICATION	REOPERATION
Particles 900 µm + NBCA	Intra-arterial	>50%	/	Wound dehiscence	No
NBCA + Particles 400 µm–700 µm	Direct percutaneous puncture and Intra-arterial	>50%	/	No	No
NBCA + Particles 400 µm	Intra-arterial	>50%	/	No	No
EVOH	Direct percutaneous puncture	>90%	/	Wound dehiscence	No
EVOH	Direct percutaneous puncture	>50%	/	No	After 6 months
EVOH + Particles 500 µm	Direct percutaneous puncture and Intra-arterial	>90%	/	No	No
EVOH	Direct percutaneous puncture	>90%	7 days	Lip asymmetry	No
EVOH	Direct percutaneous puncture	>90%	/	Cutaneous dyschromia	No
EVOH + Particles 500 µm	Direct percutaneous puncture and Intra-arterial	>90%	/	No	After 6 months
Particles 500 µm	Intra-arterial	>90%	/	No	No
Particles 500 µm	Intra-arterial	>90%	/	No	No
EVOH	Intra-arterial	>90%	2 days	No	No
EVOH	Direct percutaneous puncture	>50%	/	No	No
NBCA	Intra-arterial	<50%	/	No	No

**Table 3 jpm-13-00757-t003:** Questionnaire’s results.

SYMPTOMS	BEFORE TREATMENT	AFTER TREATMENT				
		5 COMPLETE RELIEF OF SYMPTOMS	4 MAJOR IMPROVEMENT OF SYMPTOMS	3 MINOR IMPROVEMENT OF SYMPTOMS	2 NO CHANGE IN SYMPTOMS	1 WORSENING OF SYMPTOMS
PAIN	2/14	2/2	0	0	0	0
FUNCTIONAL IMPAIRMENT	3/14	1/3	1/3	0	0	1/3
COSMETIC DEFORMITY	14/14	8/14	3/14	2/14	1/14	0
IMPAIRMENT IN DAILY LIFE	6/14	4/6	1/6	0	1/6	0
BLEEDING	4/14	4/4	0	0	0	0

## Data Availability

The data presented in this study are available on request from the corresponding author. The data are not publicly available due to patient privacy.

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
