# Peer review of "A Single-Center Retrospective Analysis of 14 Head and Neck AVMs Cases Treated with a Single-Day Combined Endovascular and Surgical Approach"

_jpm, 2023, doi:10.3390/jpm13050757_

Round 1

Reviewer 1 Report

Arteriovenous malformations (AVMs) in the head and neck are rare and still challenging for surgeons. The strategy of treatment of AVMs was personalized according to the individual features. Endovascular embolization, surgical excision and ethanol sclerotherapy are the main treatments in AVMs. In this manuscript, the authors shown us an one-stage strategy for AVMs which were treated with embolization followed on the same day by surgical resection, it could bring us new information about AVMs in the head and neck.

Author Response

We would like to take this opportunity to thank you for the effort and expertise that you contributed towards reviewing the article. 

Reviewer 2 Report

The manuscript presented describes the experience of the authors during treatment of 14 patients with arteriovenous malformations (AMV) of the head and neck region. 

The disease is rare and the diagnosis is often delayed and treatment is not standardized according to the limited experience with this disorder, showing very different presentations. 

The results of the treatment, as shown in the two patients with remarkably good results seems to support the concept of the clinical group. 

However the authors should more clearly present their concept, the differential strategy for the cases presented, be more precise in the treatment options used and give a more forward thinking perspective whow to improve the diagnosis and treatment for these group of patients. 

1) were the patients discussed in an interdisciplinary group of various disciplines experienced in vascular malformations ?

2) Give more details when ethanol sclerotherapy was used and when endovascular embolization was recommended. 

3) Regarding endovascular embolization give the indications for the type of embolic material used (see in table 2).

4) The authors report 14 patients with AVM of the head and neck region over a period of 7 years, meaning 2 patients per year were treated. How can the experience with these cases be present over the years and in the future ? Are there any efforts to establish an Italien or regional registry for AVM in general or in the head and neck region ?

5) Please give more details about the surgical intervention performed after the initial interventions by sclerotherapy or embolization. It seems that additional operations like plastic surgery or head, neck and ear reconstructions are required in some cases. 

6) The authors mention surgical interventions like cutanous and mucosal incisions. Please carify what is meant by incisions, these are also resections or just incisions to treat local edema ? 

7) In table 3 the authors show the outcome of their treatment. One patient had worsening of symptoms, although in the abstract (line 15) satisfactory results are reported for all patients. 

8) Since the number of patients is small and treatment reported expands over 7 years, are there any changes and recommendations the authors would give based on their experience over time and possible improvements observed during this period. 

9) Differential classification of the AVM is another important issue for standardized diagnosis, treatment and evaluation of outcome. Since the disease is quite rare would any national or international efforts to establish a group of experts for exchange of experience and to define recommendations or guidelines to treat these patients help ? The paper presented could also be a platform to encourage such efforts.  

10) Therefore, what is the recommendation the authors would give regarding future strategies for improvement of quality of health care for these patients ? Probably, there is need for standardization of diagnosis and treatment for AVM with different presentations and centralization in specialized centers is required ?     

Author Response

Response to Reviewer 2 Comments

Point 1: were the patients discussed in an interdisciplinary group of various disciplines experienced in vascular malformations?

Response 1: Thank you for this consideration. Yes, before each surgery we discussed the clinical case in a multidisciplinary group consisting of the surgeon, the interventional radiologist, the dermatologist, and, when children, the pediatrician. We think this note is very important, so we decided to make it more clear in the text.

Point 2: Give more details when ethanol sclerotherapy was used and when endovascular embolization was recommended.

Response 2: In our case series, ethanol sclerotherapy has never been used so we have not specified what the indications for its use are. Rather, the techniques used were percutaneous puncture and intraarterial embolization. The choice of technique was according to the anatomic structure involved and to the endovascular feature assessed with the diagnostic arteriography.

Point 3: Regarding endovascular embolization give the indications for the type of embolic material used (see in table 2).

Response 3: Thank you for pointing this out. There are no universally recognized guidelines regarding the choice of embolizing material. Often it is the expertise of the operator that makes the difference. In our case series, as specified in Materials and Methods, we preferred non-adhesive agents, such as Squid, Onix, and Phil. Sometimes procedures have been supplemented with the use of Particles to go to occlude small vessels that cannot be reached in any other way. In addition, another caution we used was the adoption of materials not containing tantalum salt in AVMs involving superficial tissues, to avoid the phenomenon of skin discoloration. Thanks to your suggestion we decide to better specify this on Material and Method.

Point 4: The authors report 14 patients with AVM of the head and neck region over a period of 7 years, meaning 2 patients per year were treated. How can the experience with these cases be present over the years and in the future? Are there any efforts to establish an Italian or regional registry for AVM in general or in the head and neck region?

Response 4: Thank you for this consideration. Unfortunately, the identification of the incidence of vascular malformations in general, and of arteriovenous forms, is difficult because of terminologies, particularly in the frequent failure to differentiate between hemangiomas and arteriovenous malformations. In 1977 Kennedy et al evaluated in a review 238 studies including more than 20 million births, the incidence of vascular malformations of all types was 1.08% on average, with a range across studies from 0.83 to 4.5%. However, this study does not allow us to determine the incidence of AVMs, which can be inferred from other analyses. According to a study by Tasnadi of 3573 children and to a study of BB Lee of 1475 cases of peripheral vascular malformations, the incidence of AVMs appears to be 12%. Other, more numerous data refer, however, to cerebrospinal AVMs, showing that the incidence of AVMs is 18 per 100000 people.

Regarding our experience, we have gradually noticed an increase in the number of patients to our attention, however, it is impossible to have national or regional data, but an effort should be made in this regard.

  • Kennedy WP: Epidemiologic aspects of the problem of congenital malformations. In: Persaud TNV

(ed) Problems of birth defects. University Park Press, Baltimore, 1977, p. 35 – 52

  • Tasnàdi G: Epidemiology and etiology of congenital vascular malformations. Semin Vasc Surg 1993.

6: 200-203

  • Lee BB: Changing concepts on vascular malformations: no longer enigma. Ann Vasc Dis 2008; 1(1):

11-19

  • Al-Shahi R, Warlow C. A systematic review of the frequency and prognosis of arteriovenous

malformations in the brain in adults. Brain 2001 124(10): 1900-1926

Point 5: Please give more details about the surgical intervention performed after the initial interventions by sclerotherapy or embolization. It seems that additional operations like plastic surgery or head, neck and ear reconstructions are required in some cases.

Response 5: That's right, each embolization was followed by surgery to remove the AVM and eventual reconstruction of the surgical area. As explained in the materials and methods, the surgery depends on the anatomical area to be treated and the defect to be reconstructed. Sometimes first intention closures were sufficient, sometimes local and locoregional flaps were used. In cases where the ear was removed, implantation of an epithesis was necessary. Of note, new surgeries were sometimes required to manage some complications, such as surgical dehiscences. However, in most cases we were able to have satisfactory results with only one surgery. Thanks to your suggestion we decide to better specify this in the text.

Point 6: The authors mention surgical interventions like cutaneous and mucosal incisions. Please clarify what is meant by incisions, these are also resections or just incisions to treat local edema?

Response 6: As clarified in the previous point, the different incisions are in relation to the different anatomical areas treated. Therefore, sometimes we made only cutaneous incisions, sometimes, for example in cases where the AVM was endoral, we made mucosal incisions. We want to specify that when we talk about incisions, we mean the type of resection chosen. We have never performed incisions to treat local edema. Thanks to your suggestion we decide to better specify this in the text.

Point 7: In table 3 the authors show the outcome of their treatment. One patient had worsening of symptoms, although in the abstract (line 15) satisfactory results are reported for all patients.

Response 7: Thank you for pointing out this inaccuracy. We actually had very good results in most patients, but not in all. We will arrange to edit the abstract.

Point 8: Since the number of patients is small and treatment reported expands over 7 years, are there any changes and recommendations the authors would give based on their experience over time and possible improvements observed during this period.

Response 8: Thank you for the opportunity to argue this point. Through our experience over the years, we have learned to manage significant complications related to this condition. This has allowed us to better select patients for treatment and improve both embolization and surgical techniques. We began to investigate alternative treatments, for example, bleomycin injection and electrochemotherapy. However, we consider our experience in this regard still too inconsistent to argue it in a scientific paper. We still consider combined treatment to be the first choice in many vascular anomalies and at present, we do not consider it outdated.

Point 9: Differential classification of the AVM is another important issue for standardized diagnosis, treatment, and evaluation of outcomes. Since the disease is quite rare would any national or international efforts to establish a group of experts for exchange of experience and to define recommendations or guidelines to treat these patients help? The paper presented could also be a platform to encourage such efforts. 

Response 9: We agree with the reviewer’s assessment. Indeed, in Italy, the SISAV (Italian Society for the Study of Vascular Anomalies) is a considerable authority. Worldwide, the ISSVA, of the eponymous classification, plays the most important role. Our study group is very active within these, and the comparison with other experts in such rare pathologies is fundamental. We would be really excited if our work would help in the management of these patients.

Point 10: Therefore, what is the recommendation the authors would give regarding future strategies for improvement of quality of health care for these patients ? Probably, there is need for standardization of diagnosis and treatment for AVM with different presentations and centralization in specialized centers is required ? 

Response 10: As with all rare diseases, we believe that the best management of these patients is in referral centers where there are established diagnostic and treatment pathways. Many patients still come to our attention after being seen in many hospitals without obtaining a definitive diagnosis or treatment. This is detrimental because since AVMs are degenerative diseases, delay in diagnosis can be fatal. Being able to evaluate so many patients allow us to better study this pathology and advance in diagnosis while always ensuring the most cutting-edge treatment. The prospects are many and many promising. What we prospect is to tailor therapies to the specific patient ensuring the best possible outcome with minimal invasiveness.

Reviewer 3 Report

Lines 127-129-Was the determination made before or after transfemoral angiography

Lines 272-278-What was the total number of patients evaluated or were the 14 patients in your study  the total number of patients you evaluated

Do you have any results longer than 6 months

Author Response

Response to Reviewer 3 Comments

Point 1: Lines 127-129-Was the determination made before or after transfemoral angiography.

Response 1: Thank you for pointing this out. This evaluation was made after the diagnostic transfemoral angiography. This is because only after this diagnostic exam we can determine the endovascular features of the AVM and choose the approach.

Point 2: Lines 272-278-What was the total number of patients evaluated or were the 14 patients in your study the total number of patients you evaluated.

Response 2: Thank you for the question. We evaluated more than 14 patients over years but only these ones fit our inclusion criteria described in the MeM section. The total number of patients was 23 but some of them doesn’t want to start the therapeutic path we proposed and some were treated with a different strategy.

Point 3: Do you have any results longer than 6 months.

Response 3: yes, most of the 14 patients of the study have been followed up for more than 6 months. For example patient 3 shown in figure 2 was treated in 2017.